# Long-Term Breast Cancer Outcomes of Pregnancy-Associated Breast Cancer (PABC) in a Prospective Cohort

**DOI:** 10.3390/cancers14194839

**Published:** 2022-10-04

**Authors:** Hyunji Jo, Seri Park, Hye Ryeon Kim, Hongsik Kim, Joohyun Hong, Jeong Eon Lee, Jonghan Yu, Byung Joo Chae, Se Kyung Lee, Jai Min Ryu, Soo-young Oh, Suk Joo Choi, Ji-Yeon Kim, Jin Seok Ahn, Young-Hyuck Im, Eun Mi Nam, Seok Jin Nam, Yeon Hee Park

**Affiliations:** 1Division of Hematology-Oncology, Department of Medicine, Samsung Medical Center, Sungkyunkwan University School of Medicine, Seoul 06351, Korea; 2Division of Hematology-Oncology, Department of Medicine, Ewha Womans University School of Medicine, Seoul 07804, Korea; 3Department of Health Sciences and Technology, SAIHST, Sungkyunkwan University School of Medicine, Seoul 06351, Korea; 4Division of Breast Surgery, Department of Surgery, Samsung Medical Center, Sungkyunkwan University School of Medicine, Seoul 06351, Korea; 5Department of Obstetrics and Gynecology, Samsung Medical Center, Sungkyunkwan University School of Medicine, Seoul 06351, Korea

**Keywords:** pregnancy, breast cancer, prospective cohort

## Abstract

**Simple Summary:**

This study explored the characteristics, treatment, and survival outcomes of patients with pregnancy-associated breast cancer (PABC) in Korea. All patients of this study received standard treatments according to the National Comprehensive Cancer Network guideline. Compared to the non-PABC group, the PABC group had a lower percentage of hormone receptor positivity, increased HER2 overexpression, and higher Ki-67 levels. No maternal complications were observed in patients with PABC. In addition, the 5-year disease-free survival and overall survival rates were significantly poorer in the PABC group than in the non-PABC group. After adjusting for tumor characteristics, PABC was still associated with poor prognosis. This is the first report of the PABC population from a prospective cohort. Exploration to elucidate biologic relevance will follow.

**Abstract:**

Background: Given that peak age of breast cancer (BC) is younger in Asians than in Western populations, relatively higher prevalence of pregnancy-associated breast cancer (PABC) has been reported. This study aimed to analyze the characteristics and clinical outcomes of PABC in Korea. Methods: We defined PABC as BC diagnosed during pregnancy or in the first postpartum year. We compared the clinicopathological characteristics and BC outcomes between patients with PABC and non-PABC patients in the prospective YBC cohort from Samsung Medical Center. Results: In total, 1492 patients were initially enrolled, and 1364 patients were included, of which 93 had PABC (6.8%). The median age of patients with PABC was 34 years. Hormone receptor expression was lower (64.6% vs 74.6%) and frequency of HER2 overexpression was higher (26.9% vs 17.6%) in patients with PABC than in non-PABC patients. The 5-year overall survival (OS) rates were 83.2% and 93.4% in patients with PABC and non-PABC patients, respectively (*p* < 0.001). The 5-year disease-free survival (DFS) rates were 72.2% and 83.8% in PABC and non-PABC patients. Conclusion: Compared to non-PABC patients, patients with PABC had poorer OS and DFS in this prospective cohort. Exploratory biomarker analysis for PABC is warranted.

## 1. Introduction

Breast cancer is the most common cancer among women worldwide. In 2020, reports estimated 2261.4 new breast cancer cases per 100,000 women globally [1]. In Korea, breast cancer is also the most common cancer in women. The incidence of breast cancer in Korea has been increasing steadily across all age groups, with a peak in the 40–49 years age group. In 2017, approximately 11% of patients were younger than 40 years old and half were younger than 50 years old at diagnosis of breast cancer [2]. The median age of patients with breast cancer in Korea and Asian countries is younger than that in Western countries; consequently, the proportion of patients with young breast cancer (YBC) is higher in Asia than in Western countries [3,4]. Although the age criterion for YBC varies from study to study, it usually refers to breast cancer occurring in patients younger than 40 years of age [5]. In addition, the aggressive subtypes are more common in YBC, and more aggressive tumor biology correlates with an increased risk of death from breast cancer [6,7,8]. Further, patients with YBC are more likely to have family history and genetic susceptibility (e.g., *BRCA* mutations) [9,10].

Considering the epidemiologic factors of breast cancer in Asia and poor outcomes of YBC, YBC has emerged as a critical health problem in Asia, including Korea. Furthermore, we hypothesized that YBC might have a unique biology rather than only being a surrogate for aggressive intrinsic subtypes. Patients with YBC require a different approach and management due to specific issues related to factors such as fertility preservation, pregnancy, work of life, and psychological problems [11,12,13]. Therefore, to satisfy the unmet need for treatment of patients with YBC, the Samsung Medical Center (SMC) launched the YBC clinic in 2013 as a representative SMC integrated care system. The YBC clinic provides special care by multidisciplinary teams, including medical oncologists, surgeons, plastic surgeons, obstetricians, gynecologists, pathologists, and radiologists. Based on the YBC clinic, the YBC prospective cohort was launched in May 2013, which included patients newly diagnosed with breast cancer under the age of 40 years or those with pregnancy-associated breast cancer (PABC).

Conventionally, PABC is defined as breast cancer diagnosed during pregnancy or within one year postpartum. Due to a continued trend of delayed childbearing, the incidence of PABC is expected to increase. Treatment of PABC is required to ensure disease control and maternal and fetal safety. Surgery and chemotherapy for breast cancer are generally safe and well-tolerated by patients during the second and third trimesters of pregnancy [14]. However, trastuzumab, endocrine treatment, and radiotherapy are contraindicated during pregnancy [14]. Therefore, treatment of breast cancer during pregnancy may be changed depending on the gestational age at diagnosis and subtypes of breast cancer. Furthermore, PABC has adverse prognostic characteristics, including larger tumor size, regional lymph node spread, and hormone receptor negativity [15,16]. Previous studies on the prognosis of PABC were controversial [17,18]. Azim et al. reported that 65 patients with breast cancer diagnosed during pregnancy had inferior disease-free survival (DFS) compared to age- and stage-matched controls (hazard ratio [HR], 2.3; 95% confidence interval [CI], 1.3–4.2) [19]. On the contrary, Litton et al. performed a case control study to determine the prognosis of patients treated with chemotherapy for breast cancer during pregnancy compared to nonpregnant patients with breast cancer. The results of the study showed similar overall survival (OS) between both groups [20]. Beadle et al. evaluated the impact of pregnancy on breast cancer in young women, and this study showed that patients diagnosed during pregnancy or within one year after delivery did not have a poorer prognosis than non-PABC patients [21].

The aim of this study was to analyze the clinicopathological characteristics and treatment and survival outcomes of patients with PABC compared to non-PABC patients in Korea, based on the YBC prospective cohort.

## 2. Methods and Patients

### 2.1. Study Design and Participants

The YBC cohort is an ongoing single-center prospective cohort study of patients diagnosed with breast cancer who are less than 40 years old or patients with PABC regardless of age. PABC was defined as breast cancer diagnosed during pregnancy or within 1 year postpartum. This study included patients who were registered in the YBC cohort between May 2013 and November 2020. Data cut-off for follow-up was 30 November 2021. In this period, a total of 2015 patients were screened for the YBC cohort. Of these, 1492 patients were prospectively registered in the YBC prospective cohort study and 128 patients were lost to follow-up. A total of 1364 patients were included in this study (Appendix A). The study participants were divided into two groups: PABC and non-PABC. The PABC group included patients with PABC. The non-PABC group included all patients except those in the PABC group. The PABC group included 93 patients diagnosed with breast cancer during pregnancy (*n* = 28) or in the first postpartum year (*n* = 65). These 93 patients were compared with 1271 women with breast cancer that was not associated with pregnancy.

### 2.2. Data Collection and Assessment

Breast cancer was diagnosed using a combination of ultrasonography, mammography, and/or magnetic resonance imaging followed by core needle biopsy. All patients, except those diagnosed with breast cancer during pregnancy, underwent imaging studies for staging work-up, including chest and abdomen–pelvis computed tomography (CT) scans, bone scan, and/or positron emission tomography. Patients diagnosed with breast cancer during pregnancy underwent only chest radiography and breast ultrasonography as staging work-up for fetal safety. However, patients who had passed the first or second trimester could undergo CT scans. All patients in the YBC cohort underwent standard treatment according to the National Comprehensive Cancer Network (NCCN) guidelines. Patients diagnosed with breast cancer during pregnancy were referred to a multidisciplinary team comprising medical oncologists, surgeons, radiologists, and obstetricians to confirm treatment plans for breast cancer and delivery.

We prospectively collected data using a predefined case-registration follow-up form, including date of diagnosis, family history of breast cancer, tumor pathologic features, all types of therapies for breast cancer (including type of surgery and use of chemotherapy, human epidermal growth factor receptor 2 [HER2] targeted therapy, endocrine therapy, and radiotherapy), and patient outcomes (including time and site of recurrence or metastasis and survival status). Hormone receptor (HR) positivity was defined as estrogen receptor (ER) and/or progesterone receptor (PR) positivity. ER and PR positivity were defined as 1% or more nuclear staining or an Allred score ranging from 3 to 8 based on immunohistochemistry (IHC) staining. HER2 status was evaluated using antibodies and/or fluorescence in situ hybridization (FISH). IHC grades 0 and 1 for HER2 were defined as negative results, whereas grade 3 was considered a positive result. HER2 amplification was confirmed by FISH in samples that received an IHC rating of 2+. The triple-negative subtype was defined by the absence of hormone receptors (ER and PR) and HER2 overexpression. The Ki-67 index was calculated as the percentage of positively stained tumor nuclear cells among all tumor cells. The Ki-67 index ranged from 0% to 100% and was defined as follows: 1+ (≤25%), 2+ (25–50%), 3+ (51–75%), and 4+ (76–100%). All patients in this cohort underwent germline *BRCA1/2* mutation testing using Sanger sequencing. Next-generation sequencing (NGS) was performed on consenting patients. For pregnant patients, gestational age was calculated from the estimated due date based on ultrasound examinations.

Each patient was classified as neoadjuvant, adjuvant, or metastatic, based on treatment settings. The neoadjuvant setting included patients who received neoadjuvant chemotherapy followed by surgery and/or adjuvant systemic treatment (chemotherapy, HER2 targeted therapy, endocrine treatment, and/or radiotherapy). The adjuvant setting included patients who underwent surgery followed by adjuvant treatment. The metastatic setting included patients with de novo stage IV disease or confirmed distant metastasis in the middle of neoadjuvant chemotherapy before surgery. PABC and non-PABC groups were compared according to clinical characteristics and treatment settings.

The primary outcomes of this study were to compare DFS and OS between the two groups. DFS was measured from the date of diagnosis of breast cancer to the date of locoregional recurrence, contralateral recurrence, distant metastasis, or death. OS was defined as the time from histologic diagnosis of breast cancer to the date of death.

### 2.3. Statistical Analysis

Descriptive statistics including median, minimum, and maximum values for continuous variables and frequencies for categorical variables were generated. Categorical variables were compared using Pearson’s chi-square test or Fisher’s exact test. Continuous variables were compared using the Student’s *t*-test. DFS and OS curves were plotted using the Kaplan−Meier method and were compared using a log-rank test. Cox regression models for multivariate analysis were used to assess the factors associated with DFS and OS. All tests were two-sided, and differences among groups were considered significant at *p* < 0.05. All statistical analyses were performed using SPSS for Windows, version 27.0 (SPSS Inc., Chicago, IL, USA).

## 3. Results

### 3.1. Patient Characteristics

A total of 1364 participants were included in the analyses. All patients of this study were female. The baseline clinicopathological characteristics of both groups are presented in Table 1. The median age at diagnosis was 34 years (range: 26–43) and 36 years (range: 19–40) in the PABC and non-PABC groups, respectively. The proportion of patients with metastatic breast cancer (MBC) was higher in the PABC group than in the non-PABC group (11.8% vs. 2.4%, *p* < 0.001). Histologically, most patients had invasive ductal carcinoma; however, lobular and mucinous types were observed in 3.4% (*n* = 43) and 3.0% (*n* = 38) of patients in the non-PABC group, respectively. There were significant between-group differences in receptor status (*p* < 0.001). Hormone receptor positivity was observed in 64.6% (*n* = 60) and 74.6% (*n* = 948) of patients in the PABC and non-PABC groups, respectively. HER2 overexpression was observed in 26.9% (*n* = 25) and 17.6% (*n* = 225) of the PABC and non-PABC groups, respectively. In total, 28.0% (*n* = 26) and 19.2% (*n* = 244) of patients in the PABC and non-PABC groups displayed the triple-negative subtype, respectively. All patients in this study underwent germline *BRCA1/2* mutation testing. Of patients in the PABC group, six (6.5%) had pathogenic *BRCA1* mutations and five (5.4%) had pathogenic *BRCA2* mutations. In the non-PABC group, pathogenic *BRCA1* and *BRCA2* mutations were observed in 51 (4.0%) and 76 patients (6.0%), respectively. There were no significant differences in pathogenic *BRCA1/2* mutations between the two groups (*p* = 0.243 for *BRCA1* mutations and *p* = 0.965 for *BRCA2* mutations).

### 3.2. Treatments

Of the treatments in the Neoadjuvant Setting of patients, who presented with clinical stages I-III, 495 received neoadjuvant chemotherapy followed by surgery and/or adjuvant therapy (Table 2). Patients in the PABC group were significantly more likely to receive neoadjuvant chemotherapy (52.7%, *n* = 49) compared to those in the non-PABC group (35.1%, *n* = 446) (*p* < 0.001). The most common regimen administered in the non-PABC group was anthracycline and cyclophosphamide (AC), followed by docetaxel (65.7%, *n* = 293). Of patients, 11 received AC neoadjuvant chemotherapy during their pregnancy, with seven and four patients receiving their first dose during the second and third trimesters, respectively. Among these patients, one was administered docetaxel after AC before delivery (Figure 1). Ondansetron was used as an antiemetic because it is classified as pregnancy category B [22] and is generally considered safe during pregnancy [23]. Of the patients diagnosed during pregnancy, four patients harboring HER2 overexpression did not receive neoadjuvant trastuzumab due to its teratogenicity [14]. These patients received trastuzumab according to the NCCN guidelines after childbirth. Mastectomy was performed in 46.9% (*n* = 23) and 41.3% (*n* = 184) of patients in the PABC and non-PABC groups, respectively. A pathological complete response was achieved in 24.5% (*n* = 12) and 31.4% (*n* = 140) of patients in the PABC and non-PABC groups, respectively (*p* = 0.335). In terms of adjuvant therapy, adjuvant chemotherapy was administered to 30.6% (*n* = 15) and 17.0% (*n* = 76) of patients in the PABC and non-PABC groups, respectively, whereas docetaxel (33.3%, *n* = 5) and capecitabine (61.8%, *n* = 47) were the most common regimens administered to the PABC and non-PABC groups, respectively. Most patients received adjuvant radiotherapy (89.8% and 92.8% in the PABC and non-PABC groups, respectively). Of the patients who received adjuvant endocrine therapy, over half of the patients in the PABC group received tamoxifen alone (59.3%, *n* = 16), followed by tamoxifen with ovarian suppression (37.0%, *n* = 10). Tamoxifen with ovarian suppression was administered to over half (51.7%, *n* = 122) of the patients in the non-PABC group.

### 3.3. Treatments in the Adjuvant Setting

In total, 827 patients received adjuvant therapy after surgery. Table 3 outlines the treatments in the adjuvant setting. There was no significant between-group difference in hormone receptor status (*p* = 0.078). However, the proportion of patients with pathological stages II and III was higher in the PABC group than in the non-PABC group (*p* = 0.037). More than half of the patients (57.6%, *n* = 19) in the PABC group underwent mastectomy with lymph node dissection, whereas more than half of the patients (59.9%, *n* = 475) in the non-PABC group underwent breast-conserving surgery. The proportion of patients who received adjuvant chemotherapy was higher in the PABC group (78.8%, *n* = 26) than in the non-PABC group (57.8%, *n* = 459). In both groups, the most frequently used chemotherapy was AC, followed by docetaxel (46.2%, *n* = 12 in the PABC group; 31.4%, *n* = 144 in the non-PABC group). Six patients diagnosed during pregnancy received adjuvant chemotherapy before delivery (AC, *n* = 4; AC followed by paclitaxel, *n* = 2). Ondansetron was also used as an antiemetic in pregnant patients. There was no significant between-group difference in the proportion of patients receiving adjuvant endocrine therapy (*p* = 0.798).

### 3.4. Treatments in the Metastatic Setting

Appendix A summarizes the treatments used in the metastatic setting. Two patients in the PABC group and eight patients in the non-PABC group experienced progression to metastatic disease during neoadjuvant chemotherapy before surgery. The median number of metastatic sites at diagnosis was three in both groups. In the PABC group, complete and partial response rates to first-line chemotherapy were 9.1% and 18.2%, respectively. In the non-PABC group, there was no complete response to first-line chemotherapy, whereas 22.6% (*n* = 7) of patients exhibited a partial response to the first-line regimen.

### 3.5. Subgroup Analysis of the PABC Group

Appendix A presents a comparison of patients diagnosed during pregnancy and within 1-year postpartum. In total, 28 women were diagnosed with breast cancer during pregnancy. The median age of women diagnosed during pregnancy was 33 years (range: 28–43), with a median gestational age at diagnosis of 20 weeks (range: 4–38). Pregnancy was discontinued in 5 (17.9%) of the 28 patients diagnosed during pregnancy (Figure 1). The median gestational age at delivery was 37 weeks (range: 30–40). Among the patients diagnosed within 1-year postpartum, the median postpartum period at diagnosis was 6 months (range: 0.25–12). There were no significant between-group differences in receptor status and treatment settings. In addition, no chemotherapy-related or fetal complications were observed.

### 3.6. Survival Outcomes

The estimated median potential follow-up times were 5.26 years (95% CI, 4.66–5.85 years) and 4.28 years (95% CI, 4.13–4.42 years) for the PABC and non-PABC groups, respectively. The estimated 5-year DFS rate was 72.2% and 83.8% in the PABC and non-PABC groups, respectively. The median DFS was not reached within the follow-up duration (log-rank *p* = 0.011, Figure 2A). During follow-up, 15 patients with PABC (16.1%) and 70 non-pregnant patients (5.5%) died. The estimated 5-year OS rates were 83.2% and 93.4% in the PABC and non-PABC groups, respectively, and the median OS time was not reached within the study period (log-rank *p* < 0.001, Figure 2B). When the PABC group was divided into pregnant and postpartum patients for survival analysis, the estimated 5-year DFS rate was 70.6% and 72.5% in pregnant and postpartum patients, respectively (log-rank *p* = 0.713, Appendix A). The estimated 5-year OS rate was 84.7% and 82.4% in pregnant and postpartum patients, respectively (log-rank *p* = 0.621, Appendix A).

To evaluate potential prognostic factors, Cox regression analysis was performed for DFS and OS. PABC, HR positivity, and a high Ki-67 index (4+, 76–100%) were identified as significant factors for DFS and OS in univariate analysis. In multivariate analysis, PABC was independently associated with shorter DFS and OS after adjustment for age, HR positivity, and a high Ki-67 index (Table 4).

## 4. Discussion

This study aimed to investigate the characteristics, treatments, and survival outcomes of patients with PABC in Korea. The current study demonstrated that, compared to the non-PABC group, the PABC group had a lower percentage of HR positivity, increased HER2 overexpression, and higher Ki-67 levels. In addition, the five-year DFS and OS rates were significantly poorer in the PABC group than in the non-PABC group. After adjusting for tumor characteristics, PABC was still associated with poor prognosis. Further, patients with PABC received standard treatments according to the NCCN guidelines, and the start of treatment was not delayed due to pregnancy. In four patients, the use of trastuzumab was restricted due to pregnancy; however, trastuzumab was commenced after childbirth. Notably, although patients with PABC had a poorer prognosis compared to non-PABC patients, most patients with PABC received appropriate breast cancer treatment.

DFS and OS were significantly poorer in the PABC group than in the non-PABC group. Studies assessing whether PABC confers a poorer prognosis than that of non-pregnancy-associated breast cancer have reported discordant findings [17,24]. Recent meta-analyses have postulated that the shorter survival of patients with PABC could be attributed to postpartum breast cancer rather than breast cancer during pregnancy [24,25]. The poorer survival of postpartum breast cancer may be explained by mammary gland involution, in which mammary epithelial cells are subsequently removed by apoptosis to return the mammary gland to its pre-pregnancy state. Involution shares numerous stromal attributes with protumor microenvironments, including immune suppression, leading to tumor growth and invasiveness [26]. In a preclinical study, the effects of involution persisted in tumor microenvironments for several years after childbirth [27]. Therefore, it has been proposed that the definition of PABC should be changed [28]. Indeed, the largest meta-analysis to date suggested that the definition of PABC should be extended to up to approximately six years postpartum [25]. In consideration of the poorer prognosis of postpartum breast cancer, no significant differences in DFS and OS were observed between pregnant and postpartum patients with breast cancer in this cohort. Since this cohort only comprised patients with YBC, more patients would be classified as having postpartum breast cancer if the postpartum period was extended. Thus, further analysis of the prognosis of PABC is warranted based on the extended definition of PABC in this cohort.

In this study, univariate analysis identified several potential prognostic factors for shorter DFS and OS. PABC, HR negativity, and elevated Ki-67 were significant factors associated with shorter DFS and OS. In multivariate analysis, PABC was significantly associated with shorter DFS and OS after adjustment for age at diagnosis, HR positivity, and elevated Ki-67. A recent meta-analysis reported that PABC was associated with poorer OS and DFS after adjustment for age at diagnosis, year of diagnosis, and tumor characteristics [19], which is consistent with the findings of this study. As MBC is a strong determinant of outcome, it was excluded from multivariate analysis.

In this cohort, the clinical stage at diagnosis was not compared between the two groups due to limitations in the clinical staging of patients with PABC. The proportion of patients that received neoadjuvant chemotherapy as initial treatment and exhibited metastatic disease at diagnosis was higher in the PABC group than in the non-PABC group. Compared to the non-PABC group, the PABC group exhibited more advanced pathological stages even in the adjuvant setting, suggesting more advanced disease at diagnosis. However, whether a higher proportion of patients receiving neoadjuvant chemotherapy directly indicates more advanced disease should be considered from several aspects. Surgery can be safely performed at any time throughout pregnancy [29] but is associated with a slightly increased risk of miscarriage [14]. In total, 11 patients who received neoadjuvant chemotherapy during pregnancy were diagnosed with PABC in the second or third trimester. The treatment plan for these patients was determined after consensus by the multidisciplinary team, and no patient received chemotherapy to delay surgery during pregnancy. As the proportion of patients with HER2 overexpression or triple-negative subtype was higher in the PABC group than in the non-PABC group, the possibility that the proportion of patients in the neoadjuvant setting would be higher in the PABC group should be taken into consideration.

PABC exhibits aggressive tumor characteristics, including advanced disease at diagnosis, nodal involvement, hormone receptor negativity, HER2 overexpression, and high histologic grade, which are associated with poor prognosis [30,31]. Similar findings were observed herein when comparing tumor characteristics between the PABC and non-PABC groups. The ratio of metastatic disease was higher and the pathological stage was more advanced in the adjuvant setting in the PABC group than in the non-PABC group. Furthermore, the PABC group exhibited lower HR positivity, increased HER2 overexpression, and elevated Ki-67 levels. Therefore, these factors may have affected the poor prognosis of PABC in this cohort. However, physiological breast changes associated with pregnancy have been reported to obscure symptoms and signs of breast cancer, leading to a delay in diagnosis and a more advanced stage at diagnosis [32]. In the largest study to date of patients with breast cancer diagnosed during pregnancy compiled retrospectively and prospectively, DFS and OS were similar between pregnant and non-pregnant patients despite differences in tumor characteristics [17]. As the prognosis of PABC has been limited to being explained solely based on stage and receptor status, recent studies have attempted to identify genomic differences in PABC. Nguyen et al. reported that breast cancer diagnosed during pregnancy exhibited an enrichment of mismatch repair deficiency mutational signature and higher frequency of mutations in the mucin gene family, which could be implicated in promoting tumor progression during pregnancy [33]. In addition, Jindal et al. recently published an analysis on the molecular profile of postpartum breast cancer. In this study, the gene expression signatures of postpartum breast cancer were consistent with increased cell cycle and T-cell activation, alongside reduced estrogen receptor signaling and TP53 activity [34]. In this regard, our cohort underwent NGS to analyze biologic tumor behaviors of PABC and YBC. Worse outcomes of PABC in the study were comparable with the recent meta-analyses results [24,25]. This result might have been from the proportion of PABC during the first postpartum year. However, there were also discordant findings to be defined, which showed poorer prognosis of all PABC patients than for the non-PABC group. One of the plausible explanations is an aggravated ER pathway caused by pregnancy, which needs to be elucidated.

Patients with YBC were more likely to have a family history and genetic susceptibility, such as *BRCA* mutations, compared to older patients [35,36]. In terms of family history of breast cancer, there was no significant difference between the PABC and non-PABC groups. All patients in this cohort were tested for *BRCA1/2* mutations, and there was no significant difference in *BRCA* mutation rate between the PABC and non-PABC groups. We, thus, speculate that *BRCA* mutations do not increase the risk of PABC. The prevalence of *BRCA* mutations in the YBC cohort was 10.1 %. The prevalence of *BRCA* mutations in young patients with breast cancer under the age of 40 years has been reported to be up to 12% [37], consistent with our data.

Systemic chemotherapy is contraindicated during the first trimester due to its teratogenicity [38]. During the second and third trimesters, chemotherapy has not been associated with significant perinatal complications [38]. Anthracycline-based regimens are recommended in pregnant patients with breast cancer, and taxanes can be used sequentially if required [14]. A recent study demonstrated that the use of taxanes in pregnancy did not increase perinatal complications or impact infant growth when compared with anthracycline-based regimens for breast cancer in pregnancy [39]. In this cohort, AC was used as the treatment of choice. Of the 28 patients diagnosed with breast cancer during pregnancy, 18 received AC before delivery and three received taxanes before delivery. No chemotherapy-related or fetal complications were observed. Ondansetron was administered as an antiemetic and was tolerated by most patients. Trastuzumab, endocrine treatment, and radiotherapy are known to be teratogenic [40,41]. Hence, these are recommended for use after delivery [14]. Four patients with HER2 overexpression started neoadjuvant AC during pregnancy and received trastuzumab after childbirth, and one patient relapsed six months after the start of adjuvant trastuzumab and died seven months later. Among the 28 patients diagnosed with breast cancer during pregnancy, only one was considered to have a poor outcome due to delayed treatment.

As compared with previous studies, this study has several strengths. First, this is a prospective study, while most studies for PABC were performed retrospectively. Other strengths are the large sample size, length of follow-up, and the standardized treatments carried out by experienced physicians and surgeons.

## 5. Conclusions

In conclusion, patients with PABC exhibited distinct clinicopathological characteristics and poor outcomes compared to young patients with breast cancer not associated with pregnancy in this cohort. Further research is warranted to investigate the biological tumor behavior of PABC and to analyze prognosis according to a newly defined concept of PABC.

## Figures and Tables

**Figure 1 cancers-14-04839-f001:**
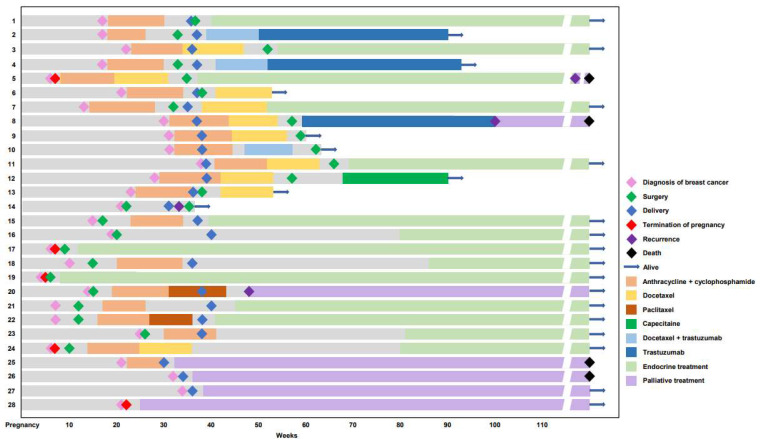
Swimmer plot of patients diagnosed with breast cancer during pregnancy (*n* = 28).

**Figure 2 cancers-14-04839-f002:**
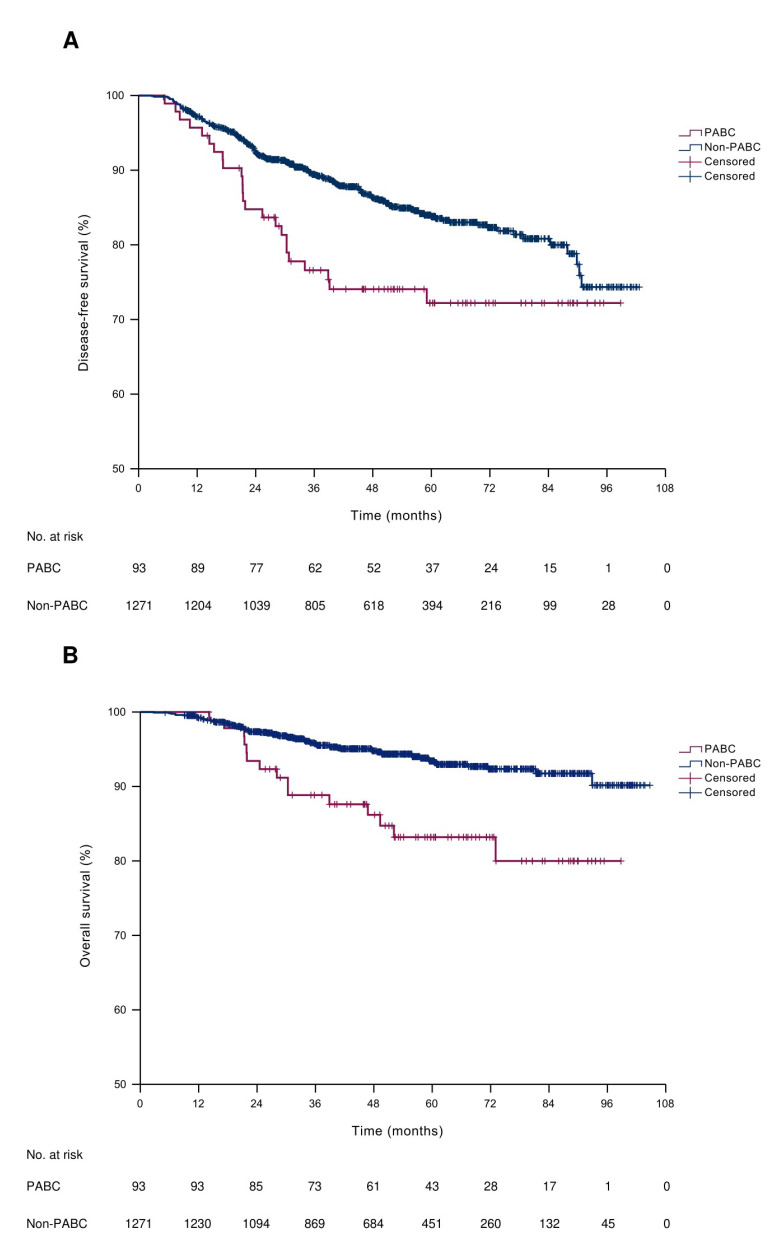
Kaplan-Meier curves of survival of the YBC cohort. (**A**) Disease-free survival. (**B**) Overall survival. PABC, pregnancy-associated breast cancer.

**Table 1 cancers-14-04839-t001:** Baseline characteristics.

Characteristics	PABC Group (*n* = 93)	Non-PABC Group (*n* = 1271)	*p* Value
Age at diagnosis (years)	34 (26–43)	36 (19–40)	0.001
Family history of breast cancer	14 (15.1)	208 (16.4)	0.775
Histology			
Ductal	87 (93.5)	1179 (92.8)	0.009 *
Lobular	1 (1.1)	43 (3.4)	
Mixed ductal/lobular	0	5 (0.4)	
Mucinous	1 (1.1)	38 (3.0)	
Others	4 (4.3)	6 (0.5)	
Receptor status			
HR+/HER2−	42 (45.2)	802 (63.1)	<0.001
HR+/HER2+	18 (19.4)	146 (11.5)	
HR−/HER2+	7 (7.5)	79 (6.1)	
Triple negative	26 (28.0)	244 (19.2)	
Ki67 semiquantitative			
1+	19 (20.4)	563 (44.3)	<0.001
2+	26 (28.0)	334 (26.3)	
3+	26 (28.0)	172 (13.5)	
4+	22 (23.7)	202 (15.9)	
BRCA1			
Pathogenic	6 (6.5)	51 (4.0)	0.243 *
VUS/Equivocal	23 (24.7)	305 (24.0)	
Wild type	64 (68.8)	915 (72.0)	
BRCA2			
Pathogenic	5 (5.4)	76 (6.0)	0.965
VUS/equivocal	22 (23.7)	327 (25.7)	
Wild type	66 (71.0)	868 (68.3)	
Treatment setting			
Neoadjuvant	49 (52.7)	446 (35.1)	<0.001
Adjuvant	33 (35.5)	794 (62.5)	
Metastatic	11 (11.8)	31 (2.4)	

Data are shown as *n* (%) per each group or median (range). * *p* value was calculated from the Fisher’s exact test. PABC, pregnancy-associated breast cancer; HR, hormone receptor; HER2, human epidermal growth factor receptor 2; VUS, variants of unknown significance.

**Table 2 cancers-14-04839-t002:** Receptor status and treatments in the neoadjuvant setting (*n* = 495).

Characteristics	PABC Group (*n* = 49)	Non-PABC Group (*n* = 446)	*p* Value
Receptor status			
HR+/HER2−	11 (22.4)	169 (37.9)	0.040
HR+/HER2+	13 (26.5)	67 (15.0)	
HR−/HER2+	6 (12.2)	56 (12.6)	
Triple negative	19 (38.8)	154 (34.5)	
Neoadjuvant chemotherapy			
AC	7 (14.3)	7 (1.6)	<0.001 *
AC + docetaxel	26 (53.1)	293 (65.7)	
AC + docetaxel + platinum	1 (2.0)	29 (6.5)	
AC + docetaxel + trastuzumab	8 (16.3)	27 (6.1)	
TCHP	4 (8.2)	66 (14.8)	
Others	3 (6.1)	24 (5.4)	
Surgery			
BCS & SLNB	21 (42.9)	177 (39.7)	0.574
BCS & ALND	5 (10.2)	71 (15.9)	
TM & SLNB	10 (20.4)	81 (18.2)	
TM & ALND	13 (26.5)	103 (23.1)	
Others	0	14 (3.1)	
pCR	12 (24.5)	140 (31.4)	0.335
Adjuvant therapy			
Adjuvant chemotherapy	15 (30.6)	76 (17.0)	0.031
Capecitabine	3 (20.0)	47 (61.8)	
Docetaxel	5 (33.3)	2 (2.6)	
Cisplatin	1 (6.7)	3 (3.9)	
Others	6 (40.0)	24 (31.6)	
Adjuvant trastuzumab	18 (36.7)	121 (27.1)	0.180
Adjuvant endocrine therapy	27 (55.1)	236 (52.9)	0.880
Tamoxifen	16 (59.3)	89 (37.7)	
Tamoxifen + goserelin	10 (37.0)	122 (51.7)	
Letrozole + goserelin	0	12 (5.1)	
Others	1 (3.7)	13 (5.5)	
Adjuvant radiotherapy	44 (89.8)	414 (92.8)	0.396 *
ypT stage			
ypT0	8 (16.3)	108 (24.2)	0.757 *
ypTis	5 (10.2)	42 (9.4)	
ypT1	19 (38.8)	165 (37.0)	
ypT2	13 (26.5)	98 (22.0)	
ypT3	4 (8.2)	33 (7.4)	
ypN stage			
ypN0	33 (67.3)	280 (62.8)	0.552 *
ypN1	8 (16.3)	102 (22.9)	
ypN2	4 (8.2)	42 (9.4)	
ypN3	4 (8.2)	22 (4.9)	
Pathological stage			
0	12 (24.5)	140 (31.4)	0.356
I	18 (36.7)	114 (25.6)	
II	10 (20.4)	114 (25.6)	
III	9 (18.4)	78 (17.5)	

Data are shown as *n* (%) per each group. * *p* value was calculated from the Fisher’s exact test. PABC, pregnancy-associated breast cancer; HR, hormone receptor; HER2, human epidermal growth factor receptor 2; AC, anthracycline and cyclophosphamide; TCHP, docetaxel, carboplatin, trastuzumab and pertuzumab; BCS, breast conserving surgery; SLNB, sentinel lymph node biopsy; ALND, axillary lymph node dissection; TM, total mastectomy; pCR, pathological complete response.

**Table 3 cancers-14-04839-t003:** Receptor status and treatments in the adjuvant setting (*n* = 827).

Characteristics	PABC Group (*n* = 33)	Non-PABC Group (*n* = 794)	*p* value
Receptor status			
HR+/HER2−	27 (81.8)	618 (77.9)	0.078 *
HR+/HER2+	2 (6.1)	74 (9.3)	
HR−/HER2+	1 (3.0)	20 (2.5)	
Triple negative	3 (9.1)	82 (10.3)	
Surgery			
BCS & SLNB	12 (36.4)	426 (53.7)	0.103 *
BCS & ALND	2 (6.1)	49 (6.2)	
TM & SLNB	9 (27.3)	210 (26.4)	
TM & ALND	10 (30.3)	101 (12.7)	
Others	0	8 (1.0)	
Adjuvant therapy			
Adjuvant chemotherapy	26 (78.8)	459 (57.8)	0.033
AC	7 (27.0)	131 (28.5)	
AC + docetaxel	12 (46.2)	144 (31.4)	
AC + weekly paclitaxel	3 (11.5)	32 (7.0)	
FAC	2 (7.7)	26 (5.7)	
TAC	0	45 (9.8)	
TC	1 (3.8)	59 (12.9)	
TCH	0	11 (2.4)	
Others	1 (3.8)	11 (2.4)	
Adjuvant trastuzumab	2 (6.1)	80 (10.1)	0.567 *
Adjuvant endocrine therapy	28 (84.8)	687 (86.5)	0.798
Tamoxifen	19 (67.9)	298 (43.4)	
Tamoxifen + goserelin	8 (28.6)	367 (53.4)	
Letrozole + goserelin	1 (3.6)	10 (1.5)	
Others	0	12 (1.7)	
Adjuvant radiotherapy	16 (48.5)	564 (71.0)	0.012
pT stage			
pT0	0	2 (0.3)	0.010 *
pTis	1 (3.0)	9 (1.1)	
pT1	11 (33.3)	458 (57.7)	
pT2	19 (57.6)	297 (37.4)	
pT3	2 (6.1)	28 (3.5)	
pN stage			
pN0	20 (60.6)	549 (69.1)	0.179 *
pN1	10 (30.3)	191 (24.1)	
pN2	1 (3.0)	42 (5.3)	
pN3	2 (6.1)	12 (1.5)	
Pathological stage			
0	1 (3.0)	11 (1.4)	0.037 *
I	9 (27.3)	397 (50.0)	
II	20 (60.6)	322 (40.6)	
III	3 (9.1)	64 (8.1)	

Data are shown as *n* (%) per each group. * *p* value was calculated from the Fisher’s exact test. PABC, pregnancy-associated breast cancer; HR, hormone receptor; HER2, human epidermal growth factor receptor 2; BCS, breast conserving surgery; SLNB, sentinel lymph node biopsy; ALND, axillary lymph node dissection; TM, total mastectomy; AC, anthracycline and cyclophosphamide; FAC, fluorouracil, anthracycline and cyclophosphamide; TAC, docetaxel, anthracycline and cyclophosphamide; TC, docetaxel and cyclophosphamide; TCH, docetaxel, carboplatin and trastuzumab.

**Table 4 cancers-14-04839-t004:** Clinical factors associated with disease-free survival and overall survival of the YBC cohort.

Variables	Disease-Free Survival						Overall Survival					
	Univariate			Multivariate			Univariate			Multivariate		
	HR	95% CI	*p* Value	HR	95% CI	*p* Value	HR	95% CI	*p* Value	HR	95% CI	*p* Value
Age at diagnosis, years	0.97	0.93–1.00	0.061	0.98	0.94–1.01	0.198	0.98	0.93–1.03	0.383	1.00	0.94–1.06	0.916
PABC	1.73	1.13–2.65	0.012	1.59	1.04–2.45	0.034	2.61	1.49–4.56	0.001	2.33	1.33–4.10	0.003
HR positivity	0.53	0.40–0.71	<0.001	0.61	0.44–0.85	0.003	0.36	0.24–0.55	<0.001	0.51	0.31–0.82	0.006
HER2 positivity	0.73	0.50–1.08	0.732				0.62	0.33–1.17	0.144			
Ki-67 4+ (76–100%)	1.77	1.27–2.45	0.001	1.31	0.90–1.89	0.162	3.05	1.96–4.77	<0.001	2.04	1.22–3.40	0.007

YBC, young breast cancer; HR, hazard ratio; CI, confidence interval; PABC, pregnancy-associated breast cancer; HR, hormone receptor; HER2, human epidermal growth factor receptor 2.

## Data Availability

The data generated in this study are available upon request from the corresponding author.

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
