# Peer review of "Long-Term Breast Cancer Outcomes of Pregnancy-Associated Breast Cancer (PABC) in a Prospective Cohort"

_cancers, 2022, doi:10.3390/cancers14194839_

Round 1
Reviewer 1 Report
Thank you for asking me to review the manuscript.
Herewith I am giving my comments. Hope these comments will be helpful for the authors.
In Statistical Analysis, you showed “Cox regression models for multivariate analysis were used to assess the factors associated with DFS and OS”. What are the variables for the adjustment variables?
What do the blanks mean in Table4?
Please move the line 297 under Table4.
Author Response
Response to Reviewer 1 Comments
Point 1: In Statistical Analysis, you showed “Cox regression models for multivariate analysis were used to assess the factors associated with DFS and OS”. What are the variables for the adjustment variables?
Response 1: Thank you for your comment.
In univariate analysis for DFS and OS, PABC, HR negativity, and elevated Ki-67 were significantly associated with wore DFS and OS. Therefore, we performed multivariate analysis with these variables. Although age at diagnosis did not show statistically significant result in univariate analysis, it was included in multivariate analysis because age is an important factor affecting survival. In summary, multivariate analysis was perfomed with age at diagnosis, PABC, HR negativity, and elevated Ki 67.
Point 2: What do the blanks mean in Table4?
Response 2: Thank you for your comment.
HER2 positivity did not show statistical significance in univariate analysis, so it was not included in multivariate analysis.
Point 3: Please move the line 297 under Table4.
Response 3: We apologize for the inconvenience. We moved the line 297 under Table 4.

Reviewer 2 Report
The manuscript entitled "Long-term breast cancer outcomes of pregnancy-associated 1 breast cancer (PABC) in a prospective cohort" by Jo et al. explored the association of different clincial characteristics, treatments and survival outcomes of PABC in a large prosepctive cohort. They identified PABC is associated with increased HER2 subtypes and poorer survival. This study is overall well designed and included a larger population to reach the conclusions. Several minor questions need to be addressed for further consdieration.
1. A more detailed comparison of results from this cohort with previous published meta-analysis need to be described in the dsicussion section, including inconsistencies and the potential reasons behind.
2. In the therapy analysis section, the author performed a deatiled analysis on different types of treatments for cancer. However, did the authors also consider the variable of pregnancy-related drugs? Some of these are hormone related and could potentially interfer with cancer-related treatments.
3. Is the metastatic-free survival also consistent with disease-free survival shown in figure2? I.e. What're the difference in terms of metastasis (sites, burden etc) between PABC and non-PABC patients?
4. Since the data collection spans 8 years and they authors need to consider the imapct of COVID-19. Does the appearance of COVID-19 influence the PABC-related survival and subtype discturbution?
Author Response
Response to Reviewer 2 Comments
Point 1: A more detailed comparison of results from this cohort with previous published meta-analysis need to be described in the dsicussion section, including inconsistencies and the potential reasons behind.
Response 1: Thank you for your comment. As your recommendation, we added the description on previous related publications in the line 379 as below:
Worse outcomes of PABC in the study is comparable with the recent meta-analyses results [24,25]. This result might be from the proportion of PABC during the first postpartum year. However, there are also discordant findings to be defined, which showed poor prognosis of all PABC than non-PABC group. One of the plausible explanations is in aggravated ER pathway caused by pregnancy, which needs to be elucidated.
Point 2: In the therapy analysis section, the author performed a deatiled analysis on different types of treatments for cancer. However, did the authors also consider the variable of pregnancy-related drugs? Some of these are hormone related and could potentially interfer with cancer-related treatments.
Response 2: Thank you for your comment.
Your point is an important issue. However, unfortunately, in this study, we did not consider the variable of pregnancy-related drugs.
Point 3: Is the metastatic-free survival also consistent with disease-free survival shown in figure2? I.e. What're the difference in terms of metastasis (sites, burden etc) between PABC and non-PABC patients?
Response 3: Thank you for your comment.
According to your comment, when metastatic-free survival was analyzed separately, there was no significant difference between the PABC group and the non-PABC group (5-year metastatic-free survival 85.1% in PABC group, 90.1% in non-PABC group, log-rank P = 0.125). Also, there was no significant difference between the two groups in terms of metastasis.
Point 4: Since the data collection spans 8 years and they authors need to consider the imapct of COVID-19. Does the appearance of COVID-19 influence the PABC-related survival and subtype discturbution?
Response 4: Thank you for your comment.
This is an important issue in the COVID-19 era. In the PABC group, there were no COVID-19-related death or delay on treatment.
